# Inhibitory Activity of Essential Oils against *Vibrio campbellii* and *Vibrio parahaemolyticus*

**DOI:** 10.3390/microorganisms8121946

**Published:** 2020-12-08

**Authors:** Xiaoting Zheng, Adam F. Feyaerts, Patrick Van Dijck, Peter Bossier

**Affiliations:** 1Laboratory of Aquaculture & Artemia Reference Center, Department of Animal Production, Faculty of Bioscience Engineering, Ghent University, 9000 Ghent, Belgium; xiaoting.zheng@ugent.be; 2Key Laboratory of South China Sea Fishery Resources Exploitation & Utilization, Ministry of Agriculture and Rural Affairs, South China Sea Fisheries Research Institute, Chinese Academy of Fishery Sciences, Guangzhou 510300, China; 3VIB-KU Leuven Center for Microbiology, 3001 Leuven, Belgium; adam.feyaerts@kuleuven.vib.be (A.F.F.); patrick.vandijck@kuleuven.vib.be (P.V.D.); 4Laboratory of Molecular Cell Biology, KU Leuven, 3001 Leuven, Belgium

**Keywords:** essential oil, *Vibrio campbellii*, *Vibrio parahaemolyticus*, volatility, quorum sensing, antibiotic

## Abstract

Vibriosis, caused by *Vibrio* strains, is an important bacterial disease and capable of causing significant high mortality in aquatic animals. Essential oils (EOs) have been considered as an alternative approach for the treatment of aquatic bacterial diseases. In this study, we evaluated the antibacterial activity of essential oils (*n* = 22) or essential oil components (EOCs, *n* = 12) against *Vibrio* strains belonging to the harveyi clade. It was verified by three different approaches, e.g., (i) a bacterial growth assay, comparing *Vibrio* growth with or without EO(C)s at various concentrations; (ii) a vapor-phase-mediated susceptibility assay, comparing the effect of EO(C)s on bacterial growth through the vapor phase; and (iii) a quorum sensing-inhibitory assay, based on specific inhibition of quorum sensing-regulated bioluminescence. The results showed that, in the bacterial growth assay, EOs of *Melaleuca alternifolia* and *Litsea citrata* at 0.0001%, *Eucalyptus citriodora* at 0.01% can inhibit the growth of *Vibrio campbellii* BB120. These EOs can also prevent the growth of *V. parahaemolyticus* strains but need to be present at a higher concentration (0.1%). Moreover, in the vapor-phase-mediated susceptibility assay, EOs of *M. alternifolia*, *L. citrata* and *E. citriodora* can inhibit the growth of *V. campbellii* BB120 through their vapor phase. However, *V. parahaemolyticus* strains (CAIM170, LMG2850 and MO904) cannot be inhibited by these EOs. Additionally, in the quorum sensing-inhibitory assay, EOs of *Mentha pulegium*, *Cuminum cyminum*, *Zingiber officinalis*, and *E. citriodora*, all at 0.001%, have quorum sensing-inhibitory activity in *V. campbellii* BB120. Taken together, our study provides substantial evidence that usage of the major components, individually or in combination, of the tested commercial EOs (extracted from *M. alternifolia*, *L. citrata*, and *E. citriodora*) could be a promising approach to control *V. campbellii* BB120.

## 1. Introduction

Vibriosis is considered an important disease hampering the aquaculture sector, resulting in serious economic losses worldwide [1]. The Gram-negative marine bacteria, *Vibrio* spp. are important aquatic pathogens and capable of causing vibriosis and several other important diseases. Interestingly, this disease, vibriosis has been reported from 48 species of aquatic animals leading to significant high mortality [2]. *Vibrio* consists of Gram-negative straight or curved rods, motile by a single polar flagellum. Moreover, several *Vibrio* strains are either obligatory or opportunistic pathogens in the marine environment globally [3,4]. Shrimps are a major marine product, with high economic value, but their commercial production has been threatened by bacterial or viral infections, especially by *Vibrio* contamination [1]; for instance, *V. parahaemolyticus* MO904 is a high-level pathogen bacterium encoding VPAHPND toxins (PirAVp/PirBVp) causing acute hepatopancreatic necrosis disease (AHPND) in shrimp [5,6]. *V. campbellii* BB120 is the causative agent of luminescent vibriosis and reported to infect brine shrimp (*Artemia franciscana*) and giant river prawn (*Macrobrachium rosenbergii*), and its virulence is likely contributed by quorum sensing regulatory gene (luxR), transmembrane transcription regulator (toxRVh), metalloprotease (vhpA), chitinase (chiA), and hemolysin (vhh) [7,8]. *V. parahaemolyticus* LMG2850 is high-level pathogenic bacterium widely associated with foodborne infection and outbreaks linked to seafood, causing vomiting and diarrhea [9], encoding the thermostable direct hemolysin-related hemolysin (trh) gene. 

It is now generally accepted that treating vibriosis or *Vibrio* contamination with antibiotics is unadvisable, as massive (mis)use of antibiotics in aquaculture lead to resistance buildup in Vibriosmaking them less effective in the long run. The excessive (mis)use of antibiotics in aquaculture also constitutes a direct threat to the environment, food safety, and even human health [10]. Therefore, alternatives to antibiotics are urgently needed. Depending on the application domain, probiotics, prebiotics, vaccines, bacteriophages, and bioactive compounds from plant extracts have been tested to control disease or avoid food contamination [11,12,13].

Essential oils (EOs) are mostly liquid, relatively volatile, and relatively hydrophobic mixtures of secondary plant metabolites; they are called EO compounds (EOCs) [14]. EOs (mixture) contain many EO components (EOCs, single compound). EOCs are bioactive molecules, mainly terpenoids and phenylpropanoids, mostly derived from intermediates of the mevalonate, methylerythritol phosphate, and shikimic acid metabolic pathways [15]. These bioactive molecules are widely used as chemical components in biology, medicine, and pharmaceutical sciences [16]. Several EOs and EOCs display a multitude of biological effects, such as bactericidal and fungicidal activity, which have been documented in in vitro studies [17,18]. Although these data are useful, the results are not directly comparable as methodologies are varying across publications. 

Diffusion, dilution, and the bioautographic techniques have been used to evaluate the antimicrobial activity of EOs in vitro [19]. However, these methods cannot unveil the volatile property of the EOs and their components, which is why a vapor-phase-mediated susceptibility (VMS) assay is included in this study [15]. This VMS assay is semi-quantitative to study the volatile characteristics of the EO(C)s, based on the Clinical and Laboratory Standards Institute (CLSI) protocol for the broth microdilution assay [20]. 

In addition, EOs are claimed to be very effective quorum sensing (QS; cell-to-cell communication) inhibitors [21]. These claims are based on experiments with a single QS molecule reporter strain. However, the outcome of such type of assay might be biased, as EO(C)s might interfere with many other physiological functions resulting in bioluminescence reduction. A specific quorum sensing-disrupting activity (A_QSI_) is measured in this study to exclude false positives. A_QSI_ is defined as the ratio between inhibition of quorum sensing-regulated luminescence in a reporter strain versus the inhibition of luminescence when the latter is independent of quorum sensing [22].

In a first step, based on bacterial growth assay (to check how EO(C)s affect the bacterial growth), vapor-phase-mediated susceptibility assay (to examine how EO(C)s act through the vapor) and specific quorum sensing-inhibitory assay (to determine how EO(C)s inhibit quorum sensing), the interference of 22 essential oils (EOs) and 12 essential oil components (EOCs) were verified against *V. campbellii* BB120. In a second step, the antimicrobial activities (bacterial growth inhibitory and vapor-phase-mediated susceptibility) of three selected EOs were examined against other members of the harveyi clade, namely *V. parahaemolyticus* CAIM170, *V. campbellii* BB120, *V. parahaemolyticus* LMG2850 and *V. parahaemolyticus* MO904. Here, for the first time, it is demonstrated that EOs of *Melaleuca alternifolia*, *Litsea citrata*, and *Eucalyptus citriodora* display antibacterial activity by inhibiting growth and quorum sensing activity of *Vibrio* strains. 

## 2. Meterials and Methods

### 2.1. Essential Oils (EOs), Essential Oil Components (EOCs), and DMSO

All EOs (*n* = 22; Appendix A) were purchased from Pranarôm International S.A. (Ghislenghien, Belgium) and all EOCs (*n* = 12; Appendix A) from Sigma-Aldrich (Steinheim, Germany). The chemical composition of all EO(C)s were characterized previously [15]. All EO(C)s were kept in brown sterile glass vials, coded to blind the experiments, and stored at 4 °C. Dimethyl sulfoxide (DMSO) was purchased from VWR International (Leuven, Belgium).

### 2.2. Vibrio Strains and Growth Conditions

*V. campbellii* wild type strain ATCC BAA-1116 (BB120) and mutant strain JAF548 pAKlux1 [22], stored in 20% sterile glycerol at −80 °C, were used in this study. The mutant strain contains a point mutant in the luxO gene, rendering the LuxO protein incapable of phosphorelay, and hence the native bioluminescence operon is not activated. In this mutant strain, upon acquisition of the pAKlux1 plasmid, luminescence becomes quorum sensing independent and hence that can be used as a control to verify if inhibition of luminescence in *V. campbellii* is specifically caused by quorum sensing (QS) inhibition. Both strains were streaked from the stock onto Luria-Bertani agar plates (Carl Roth, Karlsruhe, Germany) containing 35 g/L of sodium chloride (LB35). Subsequently, a pure colony of each strain was transferred to and cultured overnight in LB35 broth (Carl Roth, Karlsruhe, Germany) by incubation at 28 °C with continuous shaking (120 rpm).

*V. parahaemolyticus* CAIM170, *V. parahaemolyticus* LMG2850, and *V. parahaemolyticus* MO904 were also reactivated on marine agar plates (Difco Laboratories, Detroilt, MI, USA) and cultured in marine broth (Difco Laboratories, Detroilt, MI, USA) at 28 °C with shaking at 120 rpm for overnight. All bacteria cell densities were measured by spectrophotometry at 600 nm.

### 2.3. Bacterial Growth Assay

The overnight *V. campbellii* BB120 culture was re-inoculated at a dose of 10^2^ cells/mL into fresh LB35 broth, supplemented with EO(C)s individually at two concentrations (0.0001% and 0.001%) with 1% of DMSO. The control group consisted of 1% of DMSO. Then, 200 µL of the culture were put to grow in a 96-well transparent plate with a flat bottom (VWR International, Leuven, Belgium). The plate was covered with a lid and sealed with parafilm to avoid the release of the volatile EO(C)s. Later on, the plate was incubated at 28 °C with shaking for 24 h, and the cell density was monitored at 600 nm. Each concentration of EO was verified with five replicates and was determined for three independent cultures. The density of *V. campbellii* BB120 in the control group was set at 1.0, and the OD of remaining groups were normalized accordingly.

### 2.4. Vapor-Phase-Mediated Susceptibility Assay (VMS Assay)

The VMS assay was conducted as described before [15] with some modification. Briefly, *V. campbellii* BB120 was cultured and diluted with LB35 to the density of 10^2^ cells/mL. 200 µL of a 10^2^ cells/mL BB120 was added to all wells of a 96-well transparent microtiter plate with flat bottom (VWR International, Leuven, Belgium), except for wells H1 and H12 which served as blanks containing 200 µL LB35 medium. Next, 20 µL of the EO(C), without any dilution, was added on the top of the bacterial suspension in wells D/E 6-7. For each run, one microtiter plate without EO(C)s was included as a control. The microtiter plates were covered with a lid and sealed with parafilm, and then statically incubated for 24 h at 28 °C with limited air circulation. The OD value was determined spectrophotometrically at 600 nm with a Tecan Infinite 200 microplate reader (Tecan, Mechelen, Belgium) after resuspending the cells. The inhibitory vapor-phase-mediated antimicrobial activity (iVMAA) is defined as the categorized cumulative number of wells, determined by visual assessment, and excluding the volatility-center, where growth is completely inhibited. The iVMAA_90_ is defined as the inhibitory VMAA resulting in a 90% reduction of growth, in comparison to the growth of the control, as determined spectrophotometric for iVMAA. Wells in which growth was visually absent (OD_600_ ≤ 0.07) or wells with OD_600_ < 10% of OD_600_ of the control plate after correcting for the blank were counted, excluding wells to which the EO(C) was added, to determine iVMAA and iVMAA_90_, respectively. A circle enclosing the four wells to which the EO(C)s was added, was designated as the volatility-center. Around this center, concentric circles can be brawn that touch the nearest equidistant wells, with each set of wells making up a new distance category. These categories were defined to correct for the different number of wells in different categories and were ranked ordinally, with category 1 located closest to the volatility-center [15]. The resulting cumulative number of wells was classified according to the categories defined in Figure 1A and the layout shown in Figure 1B.

### 2.5. Specific Quorum Sensing-Inhibitory Assay

The specific quorum sensing-inhibitory assay was done according to by Yang et al. [22] with some modifications. *V. campbellii*, BB120 and JAF548 pAKlux1 strain, were cultured overnight and diluted to an OD_600_ of 0.1, respectively. The EOs and EOCs were supplemented at two different concentrations (0.001% and 0.0001%), and 200 µL of each culture were further incubated in 96-well white microtiter plates with flat bottom (Tecan, Mechelen, Belgium) at 28 °C with shaking. Each concentration of each EO had three replicate wells and was determined for three independent cultures. Then bioluminescence was measured after 1, 2, 3, and 4 h with a Tecan Infinite 200 microplate reader. The specific quorum sensing-inhibitory activity of the EO(C)s at a given concentration was calculated as follows:(1)$$A_{\mathrm{QSI}}=\frac{{\%\;\mathrm{Inhibition}}_{\mathrm{QS}\text{-}\mathrm{regulated}}}{{\%\;\mathrm{Inhibition}}_{\mathrm{QS}\text{-}\mathrm{independent}}}$$
with % Inhibition_QS-regulated_: percentage inhibition of QS-regulated bioluminescence in wild type *V. campbellii* BB120, % Inhibition_QS-independent_: percentage inhibition of QS-independent bioluminescence of *V. campbellii* JAF548 pAKlux1. The EO(C)s were considered as quorum sensing inhibitors if A_QSI_ was higher than 2 at one of the concentrations tested.

### 2.6. Comparison of Bacterial Growth Inhibitory and Vapor-Phase-Mediated Susceptibility of Three Selected Essential Oils against V. campbellii (BB120) and Three V. parahaemolyticus Strains (CAIM170, LMG2850 and MO904)

Three selected EOs (extracted from *Melaleuca alternifolia*, *Litsea citrata*, and *Eucalyptus citriodora*) and one inactive oil *Apium graveolens* (as a negative control), at three concentrations (0.001%, 0.01%, and 0.1%) with 1% of DMSO, were verified in four *Vibrio* strains (BB120, CAIM170, LMG2850 and MO904) following the procedure described in bacterial growth assay section with some modifications.

For the vapor-phase-mediated susceptibility assay, three selected EOs were verified against four *Vibrio* strains (BB120, CAIM170, LMG2850, and MO904) following the procedure described in VMS assay section. 

### 2.7. Statistical Analyses

Statistical analyses were performed using one-way analysis of variances followed by a Tukey’s post hoc test using the IBM statistical software Statistical Package for the Social Sciences version 22.0 (New York, NY, USA). Data were expressed as mean ± standard error. The significance level was set at *p* < 0.05.

## 3. Results

### 3.1. Essential Oils and Their Components Inhibit the Growth of V. campbellii BB120

In the first experiment, the bacterial growth inhibitory activity of 22 EOs and 12 EOCs were determined against *V. campbellii* BB120. Three of the EOs (extracted from *Cinnamomum cassia*, *M. alternifolia*, and *L. citrata*) significantly inhibited the growth of *V. campbellii* BB120 at the two doses (0.0001% and 0.001%) (Figure 2). Three of the EOCs (R-(+)-limonene, S-(−)-limonene and cinnamaldehyde) showed significant inhibition of bacterial growth of *V. campbellii* BB120 at 0.001% (50% reduction as compared to the control group, Figure 3). None of the EOCs had a significant reduction on the growth of *V. campbellii* BB120 at the concentration of 0.0001%. 

### 3.2. Essential Oils and Their Components Inhibit the Growth of V. campbellii BB120 via Their Vapor-Phase

Next, the iVMAA and iVMAA_90_ of EO(C)s against *V. campbellii* BB120 were determined to detect the vapor-phase-mediated growth inhibition of EO(C)s. The results showed that five of the EOs tested had both iVMAA and iVMAA_90_ larger than 3.0 (Table 1). They are *Artemisia herba* alba EO, with α-thujone/camphor and β-thujone, *Cinnamomum camphora* EO, rich in linalool, *M. alternifolia*, *L. citrata*, and *E. citriodora* EO. There were three of the EOCs had both iVMAA and iVMAA_90_ larger than 3.0 (Table 2). The largest inhibition activity was observed with EOC citronellal, followed by EOCs citral and α-pinene. It is worth noting that citronellal displayed a complete vapor-phase mediated antimicrobial activity (VMAA) inhibition of *V. campbellii* BB120, as growth in the whole plate was absent.

### 3.3. Essential Oils Modulate Quorum Sensing-Regulated Bioluminescence of V. campbellii BB120

Furthermore, EO(C)s were used to study the effect on quorum sensing-regulated bioluminescence of *V. campbellii* BB120. After mixing with *V. campbellii* BB120, EO of *Mentha pulegium* blocked the bacterial bioluminescence at 0.001% at 2, 3, and 4 h (Table 3). The EOs of *Cuminum cyminum*, *E. citriodora*, and *Zingiber officinalis* were observed to inhibit the bioluminescence of *V. campbellii* BB120 at 0.001% concentration for the first 2 h, afterwards, no inhibition was observed. None of the other tested EO(C)s were recorded to reduce the bioluminescence at a concentration of 0.0001% or higher concentration (Table 4). The result indicated that EO of *C. cyminum* (rich in cuminal/γ-terpinene, β-pinene and p-cymene), EO of *E. citriodora* (with citronellal), EO of *Z. officinalis* (containing α-zingiberene/β-sesquiphellandrene and camphene), and EO of *M. pulegium* (pulegone) had potential anti-QS activity on *V. campbellii* BB120. 

### 3.4. Growth Inhibitory Activity of Candidate EOs against V. campbellii BB120

The candidate EO(C)s were selected based on the following criteria: (i) in the bacterial growth assay, if the BB120 growth of each EO(C)s group was reduced by 50% relative to the control group, then the EO(C) was selected. In this case, EO of *C. cassia*, *M. alternifolia*, and *L. citrata* were selected. (ii) in the VMS assay, if both of iVMAA and iVMAA_90_ were larger than 3.0, then EO of *A. herba* alba, *C. camphora*, *E. citriodora*, *M. alternifolia*, and *L. citrata* were screened out. (iii) in the specific quorum sensing-inhibitory assay, if the A_QSI_ was higher than two, and then EO of *C. cyminum*, *M. pulegium*, *Z. officinalis*, and *E. citriodora* were the candidates. Overall, the screening results of 22 essential oils and 12 essential oil components indicated that EOs of *M. alternifolia*, *L. citrata* and *E. citriodora* were the three best candidates to inhibit the growth of *V. campbellii* BB120 in vitro (Appendix A).

### 3.5. Effect of Three Selected Essential Oils against the harveyi clade Members

The screening results of 22 essential oils and 12 essential oil components indicated that essential oils *M. alternifolia*, *L. citrata*, and *E. citriodora* were the three best candidates to control *V. campbellii* BB120 (Appendix A). Furthermore, the antimicrobial activity of the three selected EOs (extracted from *M. alternifolia*, *L. citrata*, and *E. citriodora*) was examined against four different *Vibrio* strains belonging to the *harveyi* clade (BB120, CAIM170, LMG2850, and MO904). EO of *A. graveolens* (inactive to *V. campbellii* BB120) was set as a negative control. 

The results of bacterial growth inhibitory assay at three different concentrations (0.001%, 0.01% and 0.1%) are shown in Figure 4. EOs of *M. alternifolia*, *L. citrata* and *E. citriodora* showed significant inhibition of the growth of four bacterial strains (BB120, CAIM170, LMG2850 and MO904) at 0.1%, while no significant inhibition of three *V. parahaemolyticus* strains (CAIM170, LMG2850, and MO904) was observed at 0.001% and 0.01%. The results of the vapor-phase-mediated growth-inhibitory of EOs are shown in Table 5. Surprisingly, all the selected EOs did not inhibit the growth of the three *V. parahaemolyticus* strains (CAIM170, LMG2850, and MO904) in the VMS assay (the iVMAA and iVMAA_90_ are smaller than 3.0, Table 5).

## 4. Discussion 

The study describes that among 22 essential oils (EOs) and 12 essential oil components (EOCs), EOs of *M. alternifolia*, *L. citrata*, and *E. citriodora* are considered the three best candidates to control *V. campbellii* BB120 infection. Furthermore, the study also showed that EOs (extracted from *C. cassia*, *M. alternifolia* and *L. citrata*) and EOCs (R-(+)-limonene, S-(−)-limonene and cinnamaldehyde) significantly inhibited the growth of *V. campbellii* BB120.

The main components of *C. cassia* are cinnamaldehyde and trans-p-methoxycinnamaldehyde. Cinnamaldehyde, the predominant active compound in cinnamon oil, is a natural antioxidant [23]. Several studies have shown that cinnamaldehyde can inhibit the growth of various pathogens [24,25,26]. Cinnamaldehyde contains a six-carbon aromatic phenol group. Such phenols can pass through the phospholipid bilayer of the Gram-negative bacteria cell walls and bind to porin proteins (serving as transmembrane channels for small hydrophilic solutes) to prevent the bacteria from performing their normal functions [27]. Hence, the bacterial cell membrane is considered the first target of cinnamaldehyde, altering membrane permeability, leading to loss of functional proteins and resulting in death [28]. Moreover, the antimicrobial activity of cinnamaldehyde is also attributed to the rapid depletion of the bacterial cellular adenosine triphosphate (ATP) pool [29,30] and inhibition of cell division [31].

The main components of *M. alternifolia* are terpinen-4-ol and γ-terpinene. Terpinen-4-ol is the most prominent ingredient of tea tree oil active against human and plant pathogens [32,33,34]. The favorable hydrophobic/hydrophilic character of terpinen-4-ol is thought to be the basis for antimicrobial activity, in that it is in a spot between hydrophobic and hydrophilic, which can hydrophobic enough to enter and hydrophilic enough to leave again, through the bacterial cytoplasmic membrane [35]. Furthermore, γ-terpinene, a monoterpene hydrocarbon present in tea tree oil, has antioxidant property, and this may contribute to the bactericidal activity [36].

The main components of *L. citrata* are citral and limonene. Citral and limonene are the main flavor components of citrus oils [37]. Previous studies showed that citral and limonene had appreciable antimicrobial activity against Gram-positive and Gram-negative bacteria as well as fungi [37,38,39]. The lipophilicity of citral and limonene facilitates the penetration in the lipid layers of the bacterial cell membrane and mitochondria, causing loss of their structural organization and integrity [37,40,41]. 

In the present study, neither citral nor limonene exhibited a significant growth inhibition at 0.0001%, but EO of *L. citrata* does. These results indicate that there may be a synergistic or additive antibacterial effect in the combination of citral and limonene, or other EOCs present in the EO of *L. citrata*. Some studies also concluded that EOs had greater antibacterial activity than one of their major constituents separately [42,43], suggesting that the components at a smaller percentage are critical for the antimicrobial activity. Therefore, this potential synergistic effect between citral and limonene should be investigated in more detail in the future.

Although the broth dilution assay is regarded as a standard for detecting antimicrobial activity in a liquid medium, it fails to include the volatile characteristics of the EOs and their components. Therefore, we used the vapor-phase-mediated susceptibility (VMS) assay developed by Feyaerts et al. [15], which quantifies the antimicrobial activity of a volatile on *Vibrio* in liquid culture. The VMS assay belongs to a new class of broth microdilution-based assays, where a volatile is evaluated for its biological activity in liquid culture, following its initial volatilization and migration [15]. In the VMS assay, a volatile is placed at four central wells. From there, it can spread radially symmetrical across a 96-well plate and inhibit the growth of bacteria gradually away from the volatility-center.

It is a first study to investigate growth inhibition of EO(C)s against *V. campbellii* BB120 through their vapor-phase. Among all the EO(C)s tested, the EO of *E. citriodora*, rich in citronellal (80.02%) and pure citronellal displayed the strongest inhibition activity against *V. campbellii* BB120. The citronellal from *E. citriodora* can inhibit the growth of *Candida* species; however, EOC citronellal cannot, as reported in a previous study [15]. These can partially be explained by the different enantiomers of citronellal isolated either from EOs ((S)-(−)-citronellal) or synthetic citronellal ((R)-(+)-citronellal) [44].

Quorum sensing (QS) is a cell-to-cell communication in bacteria based on secretion and detection of external signal molecules [45]. QS is involved in virulence, biofilm formation, swimming motility and bioluminescence [46]. Quorum sensing-regulated phenotypes are co-dependent on other factors and depending on the metabolic activity of the cells, potentially leading to false-positive results [47]. To address this problem, we used the specific quorum sensing-inhibitory activity A_QSI_ developed by Yang et al., as a new parameter to investigate if EO(C)s cause significant inhibition of quorum sensing-regulated bioluminescence [22]. 

*V. campbellii* BB120 contains a three-channel QS system, which is mediated by the three types of signal molecules including HAI-1, AI-2, and CAI-1 [48]. Therefore, if any reagent can prevent the accumulation of these three signal molecules or interfere with their receptors, they might block the bacterial QS-dependent virulence gene expression, making QS-disruption an interesting strategy to control bacterial disease [49]. The EOs of *C. cyminum*, *E. citriodora*, *Z. officinalis*, and *M. pulegium*, exhibited anti-QS property. The results indicated that quorum sensing might be affected by these EOs in *V. campbellii* BB120, being it in an unidentified way. EOs are mixtures, having one or a few major constituents and a variety of other minor compounds. Consequently, it is unclear which compound of the EOs is responsible at this moment.

There was no obvious anti-QS activity of cinnamaldehyde on *V. campbellii* BB120 in the present study. Our result is contradictory to the findings of Niu et al. [50], who reported that the exposure of *V. harveyi* BB886 to a concentration of 60 µM cinnamaldehyde resulted in a 55% reduction of microbial bioluminescence, and 60% of the bioluminescence of *V. harveyi* BB170 was reduced at 100 µM. This phenomenon may be explained by using a higher concentration of cinnamaldehyde to measure anti-QS activity. In our study, 0.0001% and 0.001% were used, which were equivalent to7.9 µM and 79 µM, respectively. However, these two concentrations are lower than the previously reported required concentrations (100 µM) to inhibit QS in *V. campbellii* BB120. 

Based on the screening results, the antimicrobial activity of the three selected EOs (extracted from *M. alternifolia*, *L. citrata*, and *E. citriodora*) was examined against four different *Vibrio* strains belonging to the *harveyi* clade (BB120, CAIM170, LMG2850, and MO904), and EO of *A. graveolens* (inactive to *V. campbellii* BB120) was set as a negative control. The three selected EOs were efficient to inhibit growth against *V. campbellii* BB120 but interestingly failed against all tested *V. parahaemolyticus* strains (CAIM170, LMG2850 and MO904). Although *V. campbellii* BB120 and *V. parahaemolyticus* belong to the same clade, they have their specific characteristics. For instance, it has been demonstrated that the *V. parahaemolyticus* group displays extensive genetic divergence from the *V. campbellii* BB120 group, which might be the basis for a considerably higher growth rate of *V. parahaemolyticus* relative to *V. campbellii* BB120 [51,52]. This might be a reason for the absence of activity of the selected EOs on *V. parahaemolyticus* strains. However, this needs to be investigated in more detail in the future.

Among the tested bacteria, *V. campbellii* BB120 is the most sensitive microorganism at lower concentration (0.001%) of some tested EOs, while *V. parahaemolyticus* strains require higher oils concentration (0.1%). It is assumed that many opportunistic *Vibrio* species, such as those belonging to the harveyi clade share ecological niches. Hence, any attempt to inhibit *V. campbellii* BB120 with the described EOs at low concentration might be successful but might create growth opportunity for *V. parahaemolyticus*. Therefore, applying EO can inhibit some of the harveyi clade members, but also create a growth opportunity for other members, potentially shifting problems caused by one opportunistic pathogen to another. From an ecological perspective, the potential application of EOs in aquaculture at the growth-inhibition level should be considered with great care.

EOs are complex mixtures of a wide diversity of components [14]. Therefore, their antimicrobial activity is related to their composition, configuration, amount, and their possible interaction [53]. Three different effects can be highlighted here: additive, antagonist, and synergetic [54]. The combination of clove (*Syzygium aromaticum*) and rosemary (*Rosmarinus officinalis*) EOs displayed an additive effect against the Gram-positive (*Staphylococcus epidermidis*, *S. aureus*, and *Bacillus subtilis*) and Gram-negative bacteria (*Escherichia coli*, *Proteus vulgaris*, *Pseudomonas aeruginosa*) [55]. However, a synergetic effect of this combination was observed when the mixture was applied against the human fungal pathogen *C. albicans* [55]. In addition, when this mixture was applied against the fungus *Aspergillus niger*, an antagonistic effect was exhibited [55]. 

In the present study, three selected EOs (extracted from *M. alternifolia*, *L. citrata*, and *E. citriodora*) exhibited growth inhibition against *V. campbellii* BB120 vapor-phase-mediated. The EO of *M. alternifolia* or the EO of *L. citrata* inhibited the growth of *V. campbellii* BB120, yet the EO of *E. citriodora* did not. However, the EO of *E. citriodora* inhibited quorum sensing of *V. campbellii* BB120. Even though the major components (≥10%) of these three EOs are characterized, many minor components have not been explored yet. Some studies have concluded that minor components are critical to the antibacterial activity and may contribute synergistically [56].

In conclusion, the present work represents the first attempt to study the antimicrobial effects of EOs against *Vibrio* strains belonging to the harveyi clade. EOs (extracted from *M. alternifolia*, *L. citrata*, and *E. citriodora*) display an antibacterial activity towards *V. campbellii* BB120. 

In the future, putative synergistic effects could be verified by checkerboard testing. The checkerboard testing allows determining the Fractional Inhibitory Concentration (FIC) index value. The FIC index value marks the combination of EOs that produces the largest change relative to the individual EOs minimum inhibitory concentration (MIC) [57].

## Figures and Tables

**Figure 1 microorganisms-08-01946-f001:**
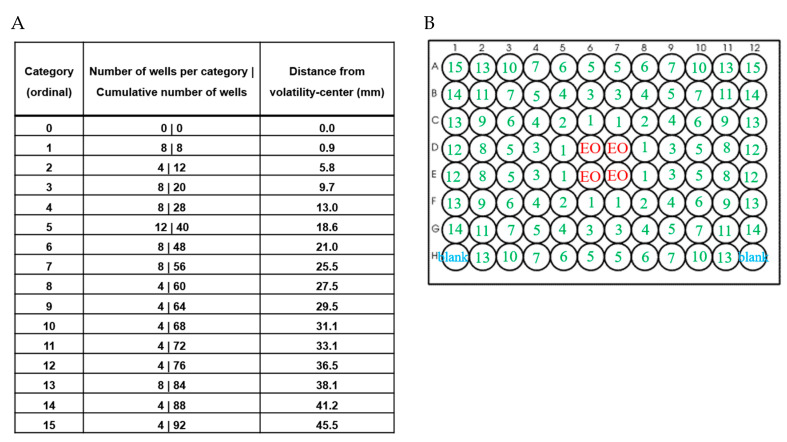
The Vapor-Phase-Mediated Susceptibility Assay (VMS) of a volatile spreads symmetrically across a 96-well plate. (**A**) The number of equidistant wells and cumulative number of wells in successive categories with their distance to the volatility-center. (**B**) The layout of the spreading of a volatile in the VMS assay under ideal conditions: the number in the wells are correspondence with each category of A [15].

**Figure 2 microorganisms-08-01946-f002:**
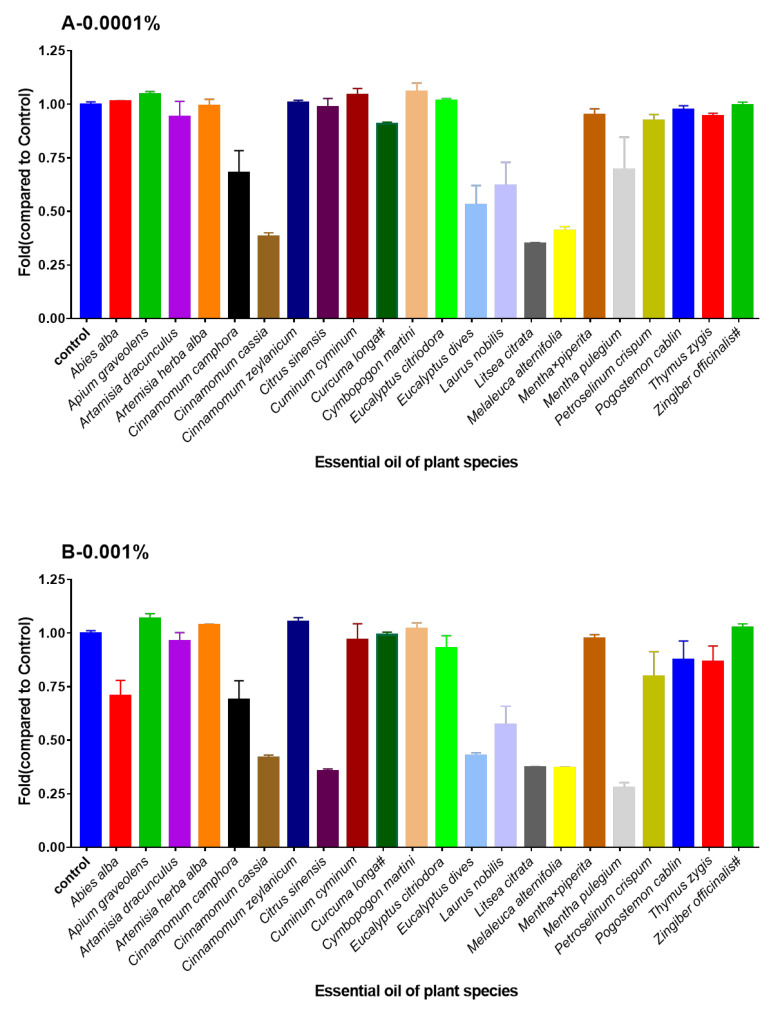
Density of *V. campbellii* BB120 at 24 h containing essential oils added at (**A**): 0.0001%, (**B**): 0.001%. The density of *V. campbellii* strain BB120 in the control group was set at 1.0, and the OD of remaining groups were normalized accordingly. If density was less than 50%, the EO was considered to have an inhibitory effect. The error bars represented the standard error of five replicates. # = organic EO.

**Figure 3 microorganisms-08-01946-f003:**
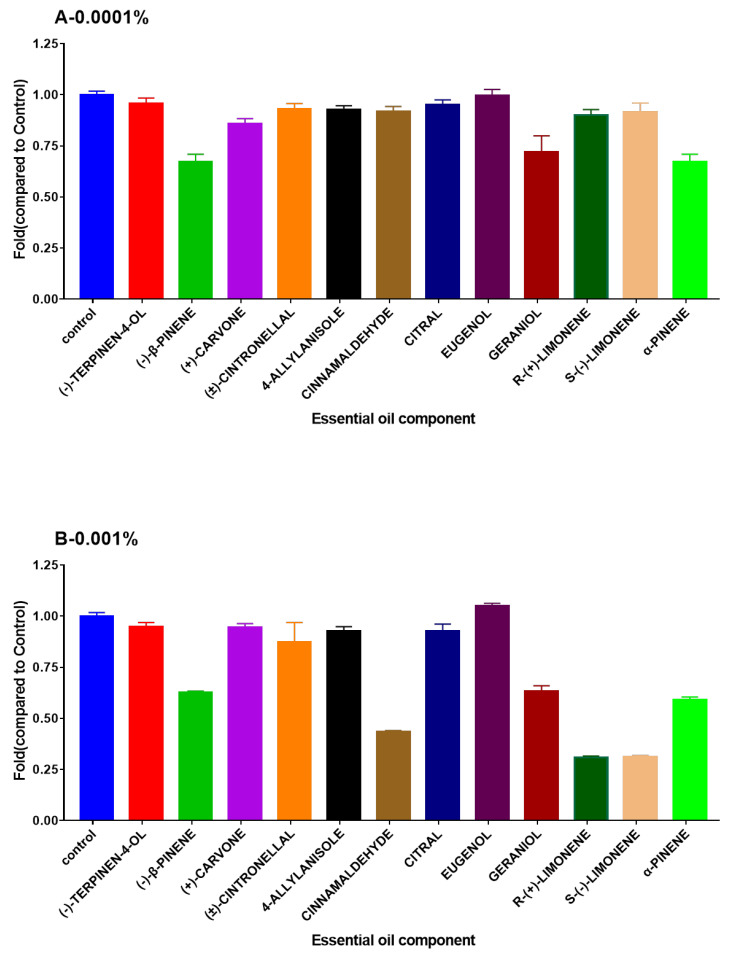
Density of *V. campbellii* BB120 at 24 h containing essential oils components added at (**A**): 0.0001%, (**B**): 0.001%. The density of *V. campbellii* strain BB120 in the control group was set at 1.0, and the OD of remaining groups were normalized accordingly. If density was less than 50%, the EOC was considered to have an inhibitory effect. The error bars represented the standard error of five replicates.

**Figure 4 microorganisms-08-01946-f004:**
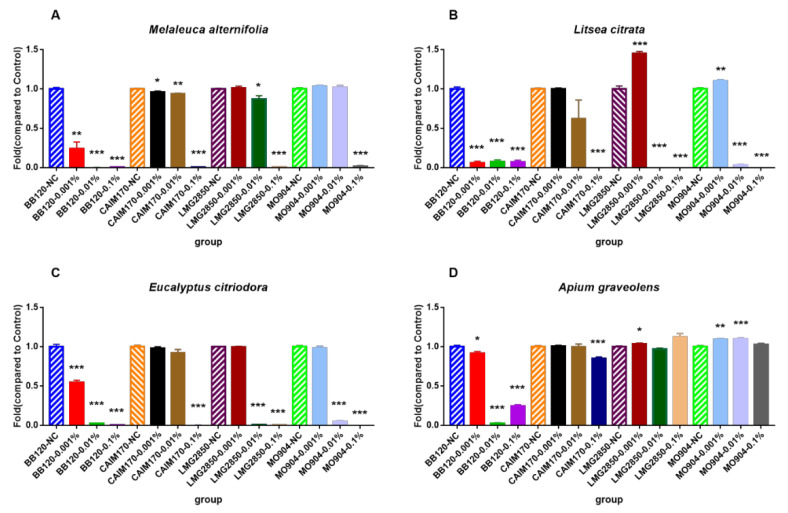
Density of four different *Vibrio* strains (BB120, CAIM170, LMG2850 and MO904) at 24 h containing essential oils added at different concentrations. (**A**): essential oil of *Melaleuca alternifolia*; (**B**): essential oil of *Litsea citrata*; (**C**): essential oil of *Eucalyptus citriodora*; (**D**): essential oil of *Apium graveolens*. The density of each control group was set at 1.0, and rest of the groups were normalized accordingly. The error bars represented the standard error of five replicates. Asterisks indicated a significant difference when compared to control (independent samples *t*-test; *: *p* < 0.05; **: *p* < 0.01, ***: *p* < 0.001).

**Table 1 microorganisms-08-01946-t001:** Comparison of iVMAA and iVMAA_90_ of 22 essential oils in testing *Vibrio campbellii* BB120. If iVMAA and iVMAA_90_ of the category were higher than 3, the EO was considered to have an inhibitory effect. The values corresponding to the inhibitory activity of *V. campbellii* BB120 was highlighted. ^#^ = organic EO.

Essential Oil of Plant Species	Number of the Wells		Category	
	iVMAA	iVMAA_90_	iVMAA	iVMAA_90_
*Abies alba*	5	5	0.5	0.5
*Apium graveolens*	0	0	0	0
*Artamisia dracunculus*	7	6	0.5	0.5
*Artemisia herba alba*	44	34	5.5	4.5
*Cinnamomum camphora*	31	26	4.5	3.5
*Cinnamomum cassia*	16	7	2.5	0.5
*Cinnamomum zeylanicum*	21	20	3.5	3
*Citrus sinensis*	0	0	0	0
*Cuminum cyminum*	19	17	2.5	2.5
*Curcuma longa* ^#^	0	0	0	0
*Cymbopogon martini* variety motia	8	8	1	1
*Eucalyptus citriodora*	92	92	15	15
*Eucalyptus dives*	17	7	2.5	0.5
*Laurus nobilis*	31	14	4.5	2.5
*Litsea citrata*	25	24	3.5	3.5
*Melaleuca alternifolia*	28	28	4	4
*Mentha × piperita*	1	0	0.5	0
*Mentha pulegium*	16	16	2.5	2.5
*Petroselinum crispum*	0	0	0	0
*Pogostemon cablin*	6	6	0.5	0.5
*Thymus zygis*	16	0	2.5	0
*Zingiber officinalis* ^#^	4	3	0.5	0.5

iVMAA: inhibitory vapor-phase-mediated antimicrobial activity (visual assessment), iVMAA_90_: iVMAA resulting in 90% reduction of growth as compared to control growth (spectrophotometric assessment).

**Table 2 microorganisms-08-01946-t002:** Comparison of iVMAA and iVMAA_90_ of 12 essential oil components towards *Vibrio campbellii* BB120. If the iVMAA and iVMAA_90_ category were higher than 3, the EOC was considered to have an inhibitory effect. The values corresponding to the inhibitory activity of *V. campbellii* BB120 were highlighted.

Component	Number of the Wells		Category	
	iVMAA	iVMAA_90_	iVMAA	iVMAA_90_
(−)-TERPINEN-4-OL	16	16	2.5	2.5
(−)-β-PINENE	0	0	0	0
(+)-CARVONE	8	8	1	1
(±)-CINTRONELLAL	92	92	15	15
4-ALLYLANISOLE	1	1	0.5	0.5
CINNAMALDEHYDE	12	12	2	2
CITRAL	44	42	5.5	5.5
EUGENOL	17	16	2.5	2.5
GERANIOL	8	8	1	1
R-(+)-LIMONENE	9	0	1.5	0
S-(−)-LIMONENE	12	5	2	0.5
α-PINENE	30	21	4.5	3.5

iVMAA: inhibitory vapor-phase-mediated antimicrobial activity (visual assessment), iVMAA_90_: iVMAA resulting in 90% reduction of growth as compared to control growth (spectrophotometric assessment).

**Table 3 microorganisms-08-01946-t003:** Comparison of the *V. campbellii* BB120 specific quorum sensing inhibitory activity (A_QSI_) of 22 essential oils (EOs) at 1 h, 2 h, 3 h and 4 h after essential oils supplementation. If A_QSI_ was higher than 2, the EO was considered to have an inhibitory effect. The values corresponding to the inhibitory activity of *V. campbellii* BB120 quorum sensing were highlighted in bold. ct = chemotype; ^#^ = organic EO.

Essential Oil of Plant Species	Special Quorum Sensing Inhibitory Activity							
	A_QSI_, 0.001%				A_QSI_, 0.0001%			
	1 h	2 h	3 h	4 h	1 h	2 h	3 h	4 h
*Abies alba*	0.75 ± 0.05 (−/−)	1.52 ± 0.35 (−/−)	0.91 ± 0.09 (−/−)	0.41 ± 0.12 (−/−)	3.93 ± 3.35 (+/+)	−1.84 ± 4.87 (−/+)	0.38 ± 6.10 (−/+)	1.66 ± 11.27 (−/+)
*Apium graveolens*	−0.41 ± 0.40 (+/−)	2.26 ± 3.32 (−/−)	0.63 ± 0.07 (−/−)	0.05 ± 0.32 (−/−)	−33.36 ± 37.24 (+/−)	1.71 ± 4.64 (+/+)	8.96 ± 20.15 (+/+)	0.98 ± 0.31 (+/+)
*Artamisia dracunculus*	0.89 ± 0.11 (−/−)	1.04 ± 0.51 (−/−)	1.81 ± 0.95 (−/−)	0.24 ± 1.78 (−/−)	−0.82 ± 9.91 (+/−)	1.70 ± 1.07 (+/+)	0.37 ± 0.12 (+/+)	0.54 ± 0.07 (+/+)
*Artemisia herba alba*	−3.98 ± 8.37 (+/−)	−3.70 ± 11.33 (+/−)	−154.84 ± 384.55 (−/+)	1.51 ± 3.69 (−/−)	8.84 ± 5.92 (+/+)	2.30 ± 9.39 (+/+)	−0.83 ± 1.52 (−/+)	−0.63 ± 2.43 (−/+)
*Cinnamomum camphora*	−0.98 ± 2.40 (+/−)	1.09 ± 0.27 (−/−)	1.13 ± 0.27 (−/−)	1.01 ± 0.04 (−/−)	−15.61 ± 32.32 (+/−)	0.05 ± 1.02 (−/−)	1.32 ± 0.13 (−/−)	1.15 ± 0.14 (−/−)
*Cinnamomum cassia*	−5.51 ± 6.86 (+/−)	0.78 ± 0.23 (+/+)	−0.03 ± 0.07 (−/+)	0.04 ± 0.05 (+/+)	7.82 ± 8.97 (+/+)	1.19 ± 0.20 (+/+)	−0.04 ± 0.14 (−/+)	0.12 ± 0.12 (+/+)
*Cinnamomum zeylanicum*	0.98 ± 0.18 (−/−)	3.24 ± 3.06 (−/−)	1.51 ± 2.45 (−/−)	2.03 ± 1.90 (−/−)	0.90 ± 4.79 (+/+)	−0.05 ± 0.66 (+/+)	−0.59 ± 0.81 (−/+)	−0.20 ± 0.50 (−/+)
*Citrus sinensis*	0.76 ± 0.04 (−/−)	1.79 ± 0.06 (−/−)	1.18 ± 0.09 (−/−)	0.07 ± 0.41 (−/−)	−10.58 ± 11.97 (+/−)	0.65 ± 0.40 (+/+)	0.29 ± 0.16 (+/+)	0.51 ± 0.22 (+/+)
*Cuminum cyminum*	0.71 ± 0.19 (−/−)	**3.25 ± 0.94 (−/−)**	1.51 ± 0.26 (−/−)	0.92 ± 0.21 (−/−)	−8.83 ± 13.36 (+/−)	0.43 ± 0.31 (+/+)	−0.47 ± 1.09 (−/+)	−0.11 ± 0.46 (−/+)
*Curcuma longa* ^#^	0.51 ± 0.15 (−/−)	−1.21 ± 1.34 (−/+)	−30.29 ± 74.95 (−/+)	−2.97 ± 7.80 (−/+)	−1.00 ± 1.12 (+/−)	0.47 ± 0.14 (+/+)	0.35 ± 0.09 (+/+)	0.69 ± 0.33 (+/+)
*Cymbopogon martini* variety motia	1.20 ± 0.04 (−/−)	1.62 ± 0.03 (−/−)	1.19 ± 0.15 (−/−)	0.85 ± 0.10 (−/−)	1.16 ± 0.36 (−/−)	24.94 ± 49.51 (−/−)	1.44 ± 2.87 (−/−)	0.62 ± 2.25 (−/−)
*Eucalyptus citriodora*	0.96 ± 0.07 (−/−)	**2.01 ± 0.62 (−/−)**	2.20 ± 3.69 (−/−)	−0.38 ± 1.71 (−/+)	−0.80 ± 1.86 (+/−)	0.05 ± 1.65 (+/+)	0.07 ± 0.72 (+/+)	0.61 ± 0.41 (+/+)
*Eucalyptus dives*	−1.10 ± 1.66 (+/−)	0.57 ± 0.69 (−/−)	0.69 ± 20.23 (−/−)	1.72 ± 4.50 (−/−)	5.57 ± 9.87 (+/+)	3.06 ± 4.13 (+/+)	−0.50 ± 0.27 (−/+)	−0.21 ± 0.21 (−/+)
*Laurus nobilis*	1.64 ± 5.84 (+/+)	6.44 ± 12.08 (−/−)	1.07 ± 0.64 (−/−)	0.57 ± 0.54 (−/−)	0.95 ± 5.87 (+/+)	0.91 ± 0.64 (+/+)	1.24 ± 2.45 (−/−)	1.29 ± 1.78 (−/−)
*Litsea citrata*	0.37 ± 0.23 (−/−)	1.97 ± 0.33 (−/−)	2.34 ± 4.13 (−/−)	0.53 ± 0.60 (+/+)	3.36 ± 22.17 (+/+)	−0.46 ± 1.63 (−/+)	−0.10 ± 0.68 (−/+)	0.49 ± 0.13 (+/+)
*Melaleuca alternifolia*	226.00 ± 560.62 (+/+)	11.03 ± 13.80 (−/−)	13.15 ± 15.05 (−/−)	2.23 ± 1.63 (−/−)	0.45 ± 8.19 (+/+)	1.31 ± 1.75 (−/−)	0.79 ± 1.05 (−/−)	0.20 ± 0.66 (−/−)
*Mentha × piperita*	1.47 ± 0.77 (−/−)	1.81 ± 0.22 (−/−)	−1.36 ± 10.91 (−/+)	2.35 ± 2.75 (−/−)	−4.92 ± 8.55 (+/−)	0.21 ± 0.76 (−/−)	−2.57 ± 4.89 (−/+)	−2.59 ± 6.39 (−/+)
*Mentha pulegium*	0.97 ± 0.03 (−/−)	**2.00 ± 0.26 (−/−)**	**2.71 ± 0.75 (−/−)**	**2.77 ± 0.83 (−/−)**	0.49 ± 0.22 (−/−)	0.89 ± 2.09 (−/−)	−1.91 ± 4.51 (−/+)	−0.95 ± 2.47 (−/+)
*Petroselinum crispum*	0.16 ± 0.05 (−/−)	−4.60 ± 10.34 (+/−)	−0.01 ± 0.78 (−/+)	−0.62 ± 1.11 (+/−)	−0.82 ± 10.46 (+/−)	0.10 ± 1.22 (+/+)	0.21 ± 0.04 (+/+)	0.41 ± 0.70 (+/+)
*Pogostemon cablin*	−0.53 ± 2.94 (+/−)	−10.86 ± 29.76 (−/+)	0.70 ± 2.66 (−/−)	0.98 ± 0.07 (−/−)	−0.57 ± 8.49 (+/−)	−0.84 ± 2.09 (−/+)	0.25 ± 2.09 (−/+)	1.98 ± 2.58 (−/−)
*Thymus zygis*	0.33 ± 0.08 (−/−)	5.43 ± 5.66 (−/−)	0.92 ± 0.12 (−/−)	0.73 ± 0.02 (−/−)	−1.26 ± 1.36 (+/−)	1.09 ± 7.15 (−/−)	−1.29 ± 2.78 (−/+)	0.62 ± 0.35 (−/−)
*Zingiber officinalis* ^#^	**2.09 ± 0.48 (−/−)**	−3.19 ± 1.84 (−/+)	12.48 ± 29.27 (−/−)	0.60 ± 5.00 (−/−)	0.96 ± 1.18 (+/+)	−1.34 ± 1.49 (+/−)	−1.08 ± 0.93 (−/+)	−0.70 ± 0.63 (−/+)

Data are mean ± standard deviation of three replicates, (+/+): Stimulation of the QS−regulated bioluminescence in *V. campbellii* BB120 and *V. campbellii* JAK548 pAKlux 1, (−/−): Inhibition of the QS−regulated bioluminescence in *V. campbellii* BB120 and *V. campbellii* JAK548 pAKlux 1, (+/−): Stimulation of the QS−regulated bioluminescence in *V. campbellii* BB120 and inhibition of the QS−regulated bioluminescence *V. campbellii* JAK548 pAKlux 1, (−/+): Inhibition of the QS−regulated bioluminescence in *V. campbellii* BB120 and stimulation of the QS−regulated bioluminescence *V. campbellii* JAK548 pAKlux 1.

**Table 4 microorganisms-08-01946-t004:** Comparison of the *V. campbellii* BB120 specific quorum sensing inhibitory activity (A_QSI_) of 12 essential oil components at 1 h, 2 h, 3 h, and 4 h after supplementation of the components. If A_QSI_ was higher than 2, the EOC was considered to have an inhibitory effect. The values corresponding to the inhibitory activity of *V. campbellii* BB120 quorum sensing were highlighted in bold. However, none of the tested EOCs were recorded to reduce the bioluminescence of *V. campbellii* BB120 at a concentration of 0.0001% or 0.001%.

Component	Special Quorum Sensing Inhibitory Activity							
	A_QSI_, 0.001%				A_QSI_, 0.0001%			
	1 h	2 h	3 h	4 h	1 h	2 h	3 h	4 h
(−)-TERPINEN−4-OL	−0.37 ± 19.7 (+/−)	0.04 ± 4.50 (+/+)	1.17 ± 2.49 (−/−)	4.65 ± 8.95 (−/−)	1.81 ± 5.65 (+/+)	0.35 ± 0.75 (+/+)	−0.30 ± 0.60 (−/+)	−0.55 ± 0.58 (−/+)
(−)-β-PINENE	0.85 ± 0.09 (−/−)	1.97 ± 0.58 (−/−)	1.07 ± 0.19 (−/−)	0.66 ± 0.17 (−/−)	−2.15 ± 13.02 (+/−)	0.65 ± 0.70 (+/+)	−7.17 ± 15.86 (−/+)	−0.39 ± 0.55 (−/+)
(+)-CARVONE	0.72 ± 0.06 (−/−)	1.05 ± 0.15 (−/−)	1.83 ± 1.10 (−/−)	−12.05 ± 37.14 (+/−)	−0.17 ± 0.30 (+/−)	3.60 ± 4.09 (+/+)	−0.01 ± 1.10 (−/+)	1.06 ± 0.17 (+/+)
(±)-CINTRONELLAL	1.11 ± 0.03 (−/−)	−1.52 ± 6.91 (−/+)	−29.45 ± 66.80 (−/+)	36.76 ± 59.34 (−/−)	−1.44 ± 1.26 (+/−)	0.68 ± 0.20 (+/+)	0.13 ± 0.26 (+/+)	0.30 ± 0.13 (+/+)
4-ALLYLANISOLE	1.09 ± 0.16 (−/−)	−0.98 ± 1.08 (−/+)	−0.58 ± 2.21 (−/+)	2.82 ± 7.45 (−/−)	2.69 ± 0.43 (+/+)	0.56 ± 0.47 (+/+)	−1.79 ± 6.14 (−/+)	0.87 ± 2.49 (−/+)
CINNAMALDEHYDE	0.74 ± 0.05 (−/−)	−5.09 ± 9.58 (−/+)	−2.05 ± 1.27 (−/+)	−0.71 ± 0.43 (−/+)	−1.42 ± 1.55 (+/−)	1.20 ± 1.13 (−/−)	1.92 ± 4.37 (−/−)	−0.21 ± 0.74 (−/+)
CITRAL	1.05 ± 0.07 (−/−)	1.69 ± 0.22 (−/−)	1.33 ± 0.51 (−/−)	0.95 ± 0.25 (−/−)	−16.14 ± 29.68 (+/−)	0.67 ± 2.21 (−/−)	1.63 ± 2.97 (−/−)	−0.80 ± 3.36 (−/+)
EUGENOL	1.08 ± 0.22 (−/−)	2.15 ± 1.17 (−/−)	0.93 ± 0.62 (−/−)	0.86 ± 0.12 (−/−)	1.59 ± 2.63 (+/+)	1.69 ± 2.99 (+/+)	1.61 ± 3.48 (−/−)	−0.02 ± 1.46 (−/+)
GERANIOL	1.31 ± 0.01 (−/−)	1.60 ± 0.03 (−/−)	1.06 ± 0.03 (−/−)	0.83 ± 0.06 (−/−)	1.65 ± 0.20 (−/−)	1.77 ± 1.06 (−/−)	0.83 ± 0.11 (−/−)	0.68 ± 0.22 (−/−)
R-(+)-LIMONENE	0.45 ± 0.07 (−/−)	1.24 ± 0.21 (−/−)	0.71 ± 0.07 (−/−)	0.57 ± 0.08 (−/−)	−3.61 ± 7.60 (+/−)	−0.29 ± 2.56 (+/−)	0.49 ± 1.16 (+/+)	−1.70 ± 4.96 (+/−)
S-(−)-LIMONENE	0.45 ± 0.06 (−/−)	1.19 ± 0.09 (−/−)	0.61 ± 0.03 (−/−)	0.50 ± 0.08 (−/−)	−2.62 ± 3.24 (+/−)	0.40 ± 0.35 (+/+)	0.16 ± 0.56 (−/−)	−0.04 ± 0.52 (−/+)
α-PINENE	0.98 ± 0.01 (−/−)	0.99 ± 0.01 (−/−)	1.07 ± 0.14 (−/−)	0.62 ± 2.49 (−/−)	−2.05 ± 1.70 (+/−)	0.68 ± 0.48 (+/+)	0.01 ± 0.97 (+/+)	0.41 ± 0.17 (+/+)

Data are mean ± standard deviation of three replicates, (+/+): Stimulation of the QS-regulated bioluminescence in *V. campbellii* BB120 and *V. campbellii* JAK548 pAKlux 1, (−/−): Inhibition of the QS-regulated bioluminescence in *V. campbellii* BB120 and *V. campbellii* JAK548 pAKlux 1, (+/−): Stimulation of the QS-regulated bioluminescence in *V. campbellii* BB120 and inhibition of the QS-regulated bioluminescence *V. campbellii* JAK548 pAKlux 1, (−/+): Inhibition of the QS-regulated bioluminescence in *V. campbellii* BB120 and stimulation of the QS-regulated bioluminescence *V. campbellii* JAK548 pAKlux 1.

**Table 5 microorganisms-08-01946-t005:** Comparison of iVMAA and iVMAA_90_ of three selected essential oils and one negative essential oil (*Apium graveolens*) in four different *Vibrio* strains (BB120, CAIM170, LMG2850 and MO904).

Bacteria and EO	Number of the Wells		Category	
	iVMAA	iVMAA_90_	iVMAA	iVMAA_90_
BB120-*Apium graveolens*	0	0	0	0
BB120-*Eucalyptus citriodora*	92	92	15	15
BB120-*Litsea citrata*	32	31	4.5	4.5
BB120-*Melaleuca alternifolia*	23	20	3.5	3
CAIM170-*Apium graveolens*	0	0	0	0
CAIM170-*Eucalyptus citriodora*	0	0	0	0
CAIM170-*Litsea citrata*	0	0	0	0
CAIM170-*Melaleuca alternifolia*	0	0	0	0
LMG2850-*Apium graveolens*	0	0	0	0
LMG2850-*Eucalyptus citriodora*	0	8	0	1
LMG2850-*Litsea citrata*	8	8	1	1
LMG2850-*Melaleuca alternifolia*	0	0	0	0
MO904-*Apium graveolens*	0	0	0	0
MO904-*Eucalyptus citriodora*	0	0	0	0
MO904-*Litsea citrata*	0	0	0	0
MO904-*Melaleuca alternifolia*	0	0	0	0

iVMAA: inhibitory vapor-phase-mediated antimicrobial activity (visual assessment), iVMAA_90_: iVMAA resulting in 90% reduction of growth as compared to control growth (spectrophotometric assessment).

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
