# Peer review of "Inhibitory Activity of Essential Oils against Vibrio campbellii and Vibrio parahaemolyticus"

_microorganisms, 2020, doi:10.3390/microorganisms8121946_

Round 1

Reviewer 1 Report

The study by Zheng et al., assesses a broad suite of essential oils and essential oil compounds against V. campbelli and some V. parahaemolyticus strains. I think it is a wide study that provides insights on the activity of these EO(C)s and also among different approaches. Volatility assay is something new that is based on an interesting concept and that along with the EO(C) results could be useful contribution to the literature. However, there are some points that certainly need closer attention. I am quite skeptical on the AQSI assay that tries to evaluate any impact on the oils to the QS systems. I think that some of the results corroborate to the previously expressed concerns on this assay as per its replicability and accuracy. Also, I would suggest that the authors build further the background on how previous research has failed on that assessment, what would be the role and the reason studying this aspect of QS using antimicrobial compounds, what would be expected from this research and how that might be translated to implementation. Toxicity of the essential oils for the cultured organisms is something that has been omitted in the manuscript and could lead to a dead end in the application of such treatments. Last, I would recommend that the discussion be more results-focused.

Title: vibrios instead of Vibrio’s

Title: I would suggest that you more specifically revise the title eg. Inhibitory activity of essential oils against V. campbellii and V. parahaemolyticus.

Abstract: Make sure that all bacterial species names are italicized through the entire manuscript.

Line 51: I agree with the point against antibiotics but I would recommend revising the ¨pose a danger to aquatic animals¨ because I think it is an overstretch. Consider adding one more sentence in order to briefly explain what are the major side effects and problems that can be caused by the excessive administration of antibiotics.

Line 54: bacteriophages is also a quite potent alternative to antibiotics that has increasingly been gaining interest lately.

Lines 56-59: Consider rephrasing this part so that it connects better with the previous and next sentence. I am not sure that the meaning here is very clear.

Line 81: What was the reason for omitting the AQSI during the second step?

Line 95: please provide citation for the mutant strain so that its reliability and functionality are confirmed. Also, it would be quite useful to provide information on how the mutant strain functions, what is the role of this plasmid etc.

Line 121: please define concentration of EO(C)s

Figure1: I think that category 4 has 8 cells, category 5 has 12 and cat 7 has 8. Please take a second look at it as well.

Line 147: Maybe a little bit of background or/and citations on how BB120 assay works, would be useful.

Line 153: ¨… oils against V. campbellii and three V. parahaemolyticus strains¨. Since they are not so many different species you could as well name them.

Line 189: I think I count 16 in table 1 but probably you refer to the category in table 1. It is though unclear how the calculation of category numbers was done. Could you please explain that in materials and methods?

Line 190: I think ¨among these¨ shall be rephrase because you mention all 5 here.

Line 211: I would like to see some further background on why QS is important to be tested here and a little bit more info on previously relevant reports. I think you cited one research in the introduction but I would suggest that you further elaborate there about the value that new results may add. In addition, I think that the BB120 assay target only the AI-2 QS signal which is just a small fraction of the QS pathways that vibrios entail. It would be nice to address such potential limitations in the text as well. May I assume that these limitations prevented you from using that method during stage 2 when you tested V. parahaemolyticus ?

Line 234: Are the chosen thresholds based on some specific logic or were they arbitrary?

Excuse my tenacity but I have some further questions for these tables. In table 3 there are some values that seem significantly off, with quite large standard errors. Wouldn’t you think that such findings might hamper the credibility of the assay? Let me also complement that there are reports in the literature but I can tell from personal experience that the bioluminescent assays of BB120 can be quite precarious and not always repetitive.

In tables 3 and 4 there are some minuses (-) in some of the cells before the numbers as well. Are these typos? Also, what is the essential meaning of the parenthesis (+,-) or (+,+) or etc.? Wasn’t the purpose of A_QSI to explain and correct any false positive results?

Line 245: the subtitle of 3.4. is almost the same as 3.1. I guess you mean something like a summary of 3.1, 3.2 and 3.3 that led to the final candidates that were used to test V. parahaemolyticus, right?

Line 250: L. citrata

Line 254: I think the selection of the final candidates by the authors confirm my concern on how important role should the AQSI results play during selecting ideal amc candidates. If the authors share similar concerns, I suggest that they explicitly state them in the manuscript.

Also, consider moving table 5 to the supplementary information.

Line 272: It is quite positive that the EOs were effective against another Vibrio species, however, there might be a red flag here. I think that an issue rising here is that such concentrations must be assessed in terms of toxicity against the cultured organisms that are going to consume the EO-treated feed. As the volatility assay indicated, the inhibition is probably not due to the EO vapours.

Line 333: The effect that volatile compounds may have on bacteria is a very nice idea, however, its implications for potential applications could be addressed too. For instance, I imagine that this is not a good candidate if EO-treated feed were to be produced simply because they would be ineffective very soon due to evaporation.

Line 341: as previously mentioned, QS is a rather complicated process and some further background on it should be built. In addition, there are plenty of signaling molecules (autoinducers) and unfortunately, there is not a single assay that can test all of them. The authors correctly mention some of them but I am certainly not sure that the AQSI assay is either appropriate or broad enough to test them all. Correct me if I am wrong, but I think that BB120 can respond only to AI-1 and AI-2 autonducer levels.

Line 342: this could be moved to materials and methods.

Lines 363-380: in these lines there are some useful information on the bacterial strains, potentially better fit for the introduction to articulate better why going after them. However, I think they do not contribute much to the discussion part or explain any of the results.

Line 398: Have you thought of using the candidates to evaluate a potential synergistic effect?

Reviewer 2 Report

The present MS by Zheng et al. provides an exciting approach to bacterial growth inhibition by single essential oils, their combinations, or a panel of derived components using three approaches. The study is elegantly combined variated techniques, treatments, and bacterial strains. This multivariate analysis partially clarifies the natural components' controversial topic as a bactericidal element, while the scientific community takes advantage of the real scope and power of medicinal plants" use and application. Thus, the study is novel and timely. However, before suggesting acceptance of the same, the following concerns should be discussed.

Along with the text, the name citrata and citrate are used interchangeably as a single species of Litsea. Whatever right, the genus Litsea, which comprises 622 known species, presents a synonymic controversy on the particular variety Litsea cubeba. L. cubeba is also known as Litsea citrata (Blume) 10.1080/10412905.2019.1611671. Please provide a brief clarification of the taxonomic status.

In general, fish and shellfish aquaculture suffer from vibriosis as a significant cause of diseases. In addition to the Harveyi clade, several other Vibrio species are of significant concern in fish, Eg. V. anguillarum, and V. ordalii. Do authors expect these results could be extended to fight against these species as well? Please provide a short comment.

Resulting from in vitro assays, the authors propose using the VMS assay as an antimicrobial control method in aquaculture. Feyaerts et al., reported positive effects on fungi, and here the authors did it so with bacteria suggesting a wide antimicrobial spectrum. In vivo using mammalian models, applying EO(C) has been observed effectively in fighting respiratory diseases. The authors suggest the practical applicability of the same in aquaculture. However, the application on any aquatic animal seems complicated. Please provide some technical advice on this issue. Moreover, performing a short trial where the VMS is supplied in bubbling air on Artemia tanks should enormously enhance and fully support the present findings.

Cinnamaldehyde, the predominant phenolic active compound present in cinnamon oil derived from the three species used in this study, is a natural antioxidant that can pass through the phospholipid bilayer of the vibrio cell walls and disrupt the physiological responses of the same. Here, a robust inhibitory activity was only observed for C. cassia. Please provide some speculation of why the other two species failed to provide a strong inhibitory effect.

Quorum sensing was affected by some of the EO(C). However, the effect of L. citrate and M. alterifolia on this communication mechanism is null, suggesting that only a direct lytic effect could be attributed. Do authors have some hypotheses on the action of the EO(C) on the electrical mechanism disruption? Eg., the long-range electrical signaling mediated by ion-channels 10.1038/nature15709

The lack of association between the Vibrio strain numbers and their scientific names complicates the correlation with the results.

L391-393, EO, or EOC have been regarded as a non-specific antimicrobial. However, here, the authors highlight a major constrain of using these particular compounds. Moreover, the EO acts only on some specific species in a particular administration-specific manner. How to overcome it?

L44, add a short sentence emphasizing the particular author's interest in crustacean vibriosis.

L420, spell out "MI", minimum inhibitory concentration

Table 1, Cinnamomum zeylanicum is presented highlighted in the table but not described in the results section.

Italics are missing for all scientific names.

Reference (9), "Carbone D, Faggio C" is dubious by this referee. This referee suggests substituting it for more reliable papers like 10.3389/fphys.2019.00785 and 10.3389/fimmu.2015.00512

Round 2

Reviewer 1 Report

I would like to thank the authors for addressing all my comments, providing thorough replies with valid arguments.

I think that the quality and the results of the manuscript would be a valuable contribution to the current literature.

One last thing I would like to recommend is that the authors consider including a statement in the discussion on how unstable the use of bioluminescent strains-based assays can be. I believe it would save time and make people who intend to use such assays more aware of what to expect. This is a personal opinion that was also acknowledged by the authors in their letter, hence I am leaving this as an open suggestion, up to the authors, but it would potentially help more fellow researchers.

I am happy to consent to publication.